# Mixed methods exploration of Ghanaian women's domestic work, childcare and effects on their mental health

**Nkechi S. Owoo**[☯], **Monica P. Lambon-Quayefio**[ID]*[☯]

Department of Economics, University of Ghana, Accra, Ghana

☯ These authors contributed equally to this work.
* mplambon-quayefio@ug.edu.gh

## Abstract

This research paper aims to understand the effects of time spent in domestic work, including childcare, on women's mental health in Ghana. The paper adopted a triangulation convergence mixed methods approach. The quantitative information was sourced from two waves (2009/ 2014) of the Ghana Socioeconomic Panel Survey (GSEPS) while qualitative information was obtained from in-depth interviews with couples and key informants from five (5) regions, representing diverse ethnic backgrounds, in Ghana. Employing fixed effects regressions and a multinomial logistic regression model with fixed effects, we find that domestic work contributes to poorer mental health outcomes among women. These results are consistent, even when we correct for potential self-selectivity of women into domestic work. We also examine whether the relationship is differentiated between women of higher and lower socioeconomic status. We find that women from wealthier households who spend increasing time in domestic work have higher odds of mental distress. These results are supported by the qualitative data- women indicate increasing stress levels from domestic work and while some husbands acknowledge the situation of their overburdened wives and make attempts, however minor, to help, others cite social norms and cultural expectations that act as a deterrent to men's assistance with domestic work. Efforts should be made to lessen the effects of social and cultural norms which continue to encourage gendered distributions of domestic work. This may be done through increased education, sensitization and general re-socialization of both men and women about the need for more egalitarian divisions of household work.

## I. Introduction

A plethora of research has been conducted on health effects of paid work in both developed and developing country settings. Broadly, some studies show that employment has positive implications for both physical wellbeing [1] and general mental health [2]. Other studies on paid work and health have discovered that beyond mere employment, the nature and quality

**Data Availability Statement:** The authors do not have the right to share the quantitative data. This is because the second wave of the quantitative data has not been made publicly available yet by the

principal investigator responsible for the data. The authors obtained the second wave of the panel data directly from the principal investigator. Access to this data can be made by directly emailing the principal investigator at rdosei@yahoo.co.uk. However, the first wave of the data is publicly available at https://microdata.worldbank.org/index. php/catalog/2534 on the World Bank LSMS data. Also, the authors are unable to share the qualitative data due to a clause in the contract of the organisation that funded the collection of the data for a different project that the authors are involved in. The clause enjoins that the qualitative data is made available only after the completion of the project. As such, the authors will be breaching this clause if the data is made available now. However, upon completion of the project, the data will be made publicly available on this website (https://g2lm-lic.iza.org/) or by directly contacting the co-author on (mplambon-quayefio@ug.edu.gh).

**Funding:** The author(s) received no specific funding for this work.

**Competing interests:** The authors have declared that no competing interests exist.

of work is a key factor in the paid work-health relationship. For instance, low-quality or poorly paid jobs are associated with unfavorable effects on health [3, 4].

Despite the extensive focus on paid work, of equal importance and yet less attention, is research focus on unpaid domestic work and its implications for women's health and wellbeing. It was not until the 1980s, coinciding with feminist social movements, that attention was drawn to women's responsibilities within the home environment. While paid work is often perceived as a source of status, power and opportunities, domestic work, despite being essential for well-being and survival, is socially undervalued, unpaid and disregarded. Another reason for the lower research interest is the dearth of data on time-use, particularly in developing country settings. Despite the debates of the 1980s, it was not until the 1990s that scientific inquiry began into this subject area. Even less studied are the health repercussions of domestic work on women, the primary performers of these activities. Although research, predominantly in developed countries, has noted potential health effects of domestic work [5, 6], these issues have not been explored in many developing country settings. An exception is the study by [7] which examined the association between housework overload and common mental disorders among women in urban Brazil. They found that housework overload, calculated as the weighted number of domestic activities that women engaged in, had adverse effects on their health.

In many developing countries, including Ghana, societal norms and expectations have created a gendered division of labour, with women bearing the primary responsibility for household and domestic work while men play a primary role in paid work and as breadwinners. In situations where women also participate in the labour force, this creates a double burden of work. According to statistics from the 2015 Ghana Labour Force Survey, close to 74% of women are in the labour force, with 65% being currently employed. Despite their labor force participation, women continue to shoulder a greater burden of household responsibilities, as indicated in Fig 1 below. On average, women in Ghana spend 4 times as much time in domestic work and secondary childcare, compared to their male partners. It has been suggested that this unbalanced division of housework is one of the factors that generally contributes to the observed adverse health differences between men and women, including psychological distress

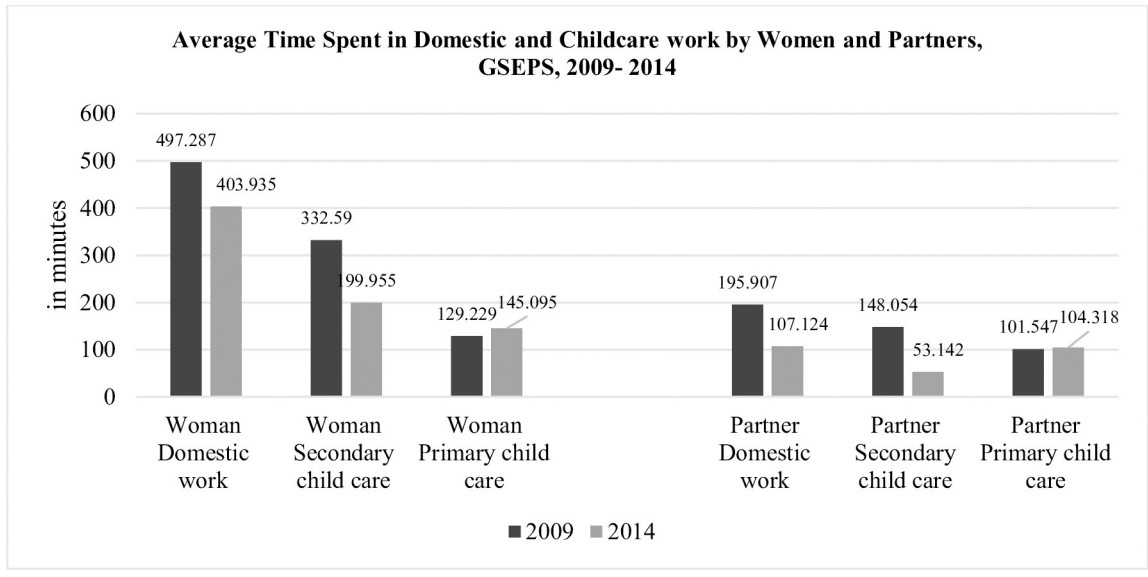

**Fig 1. Distribution of domestic and childcare work between Ghanaian couples, GSEPS, 2009–2014.** Author Computations, GSEPS.

[8] and depression symptoms [5]. In Ghana, the double burden of home and labour market responsibilities is, therefore, likely to take a toll on women's mental health.

The role overload theory is commonly invoked to explain health effects of domestic work among women. On the one hand, women's involvement in multiple roles (such as home-maker, labour force participant, community contributor) may lead to the creation of numerous avenues for achievement and intrinsically satisfying outcomes. Studies have suggested that women have better health from complementing domestic work with market participation as the job environment helps to build women's self-esteem and confidence in decision-making, provide social support for otherwise isolated individuals, and also encourage experiences that would enhance her general satisfaction. Depression may be alleviated if setbacks in one domain can be tempered with achievements in another [9–11]. Additionally, research suggests that wives in general benefit from employment because it adds a culturally- valuable, significant worth to their roles [12].

Although involvement in domestic work may provide subjective satisfaction to women and autonomy in the pace and order of house tasks, domestic work does not confer any institutional recognition to women and is not a source of social prestige. Combined with the arduous and time-consuming nature of these activities, particularly in developing country settings where domestic technology may not be as readily available as in developed countries, these activities are likely to be depression-prone [13–15]. The combination of different roles may, therefore, contribute to greater stress and anxiety due to the absolute time and energy demands of these multiple obligations [16]. Often referred to as role conflict, mothers of young children may be overburdened by the demands of childcare, combined with the demands of labour work; and wives may display more depression symptoms because they generally receive fewer role-related rewards and more role-related stresses [17]. Therefore, when women's total domestic workload is high, the combination of different roles may actually damage her health. Furthermore, the influence of social economic status may moderate the relationship. On the one hand, women with higher socioeconomic status may experience lower mental distress if they are better able to afford domestic appliances and domestic helps to reduce time spent in domestic work. On the other hand, wealthier women who spend more time in domestic work may have poorer mental health outcomes if this increases the opportunity costs of earnings and/or leisure activities, among other reasons.

Although some researchers have forwarded the argument that performing housework is synonymous with physical activity and therefore better health [1, 18, 19] found that while increasing duration of physical exercise and brisk walking indeed decreased resting pulse and BMI, no such trends were observed with heavy housework. [20] also found, in their multinational study, that the least favourable health outcomes were observed among women with greater involvement in the labour market and also high household hours. [21] also found that although activities like running, swimming and gym workouts performed during their leisure time helped women to stay mentally healthy, housework did not generate the same mental health benefits and shows no protective effects against symptoms of depression. It is important to emphasize that studies that have found that domestic work has positive implications for women's health have often examined the evidence for older women who no longer work and therefore may not have the same time constraints of combining paid and unpaid/household work [6, 22, 23].

This study, therefore, aims to explore effects of women's domestic burdens on their mental health in Ghana. Furthermore, we examine whether the strength of the association between housework and mental health (if any) is affected by the household's wealth status. The study also presents a more refined methodological approach to the examination of these relationships. An important limitation of existing studies on work and health is the potential for self-

selectivity, where comparatively healthier individuals are more likely to enter the workforce or engage in domestic activities [24, 25]. Neglection of this selectivity issue could lead to biased regression estimates. This is taken into account in this research through the adoption of a propensity score matching approach.

## II. Data and methodology

This research paper relied on the use of a mixed methods approach in the examination of the relationship between time spent in domestic work by women and their associated mental health outcomes. Mixed methods research has a long history in the social sciences and has recently gained wide acknowledgment as a means of combining both quantitative and qualitative methods for data analysis in a single study. Of the various approaches to mixed methods research, the triangulation convergence mixed methods approach is adopted in this study, in order to draw valid conclusions about the relationships of interest. Each dataset is described in greater detail below.

### a. Quantitative data

The data used for the quantitative analyses is the Ghana Socioeconomic Panel Survey (GSEPS). Two waves of the data are available- the first wave of data collection took place over a 6-month period (November 2009 to April 2010); the second wave started in 2014 and was completed in 2015. The Ghana Socioeconomic Panel Survey is a joint effort between the Economic Growth Centre (EGC) at Yale University and the Institute of Statistical, Social and Economic Research (ISSER), at the University of Ghana (Legon, Ghana). The survey is principally funded by the EGC, designed by both the EGC and ISSER, and carried out and supervised by ISSER. Technical support is provided by the University of Michigan's Survey Research Center International Unit (SRC IU). Although wave 1 is publicly available (https://microdata.worldbank.org/index.php/catalog/2534), Wave 2 data is available only upon request. Interested researchers are asked to provide information about themselves and the use to which they will put the data before access is granted.

The survey provides regionally representative data for the then 10 regions of Ghana. In total, 5010 households from 334 Enumeration Areas (EAs) were sampled. Fifteen households were then selected from each of the EAs. The number of EAs for each region was proportionately allocated based on the estimated 2009 population share for each region. EAs for Upper East Region and Upper West Region, which have relatively smaller population sizes, were over-sampled to allow for a reasonable number of households to be interviewed in these regions. Analysis of the quantitative data was carried out in STATA version 15.0.

**i. Selected variables of interest (mental health and time in domestic work).** In the GSEPS data, women's mental health is measured by the Kessler Psychological Distress Scale (K10). This is a well-validated, extremely beneficial clinical measure of psychological symptoms well-known for its ease of use, accessibility, high predictability, and high factorial and construct validity [26]. The measure can be used as a brief screen to identify levels of distress in both developed [27, 28], and developing country contexts [29, 30].

The K10 scale involves 10 questions about emotional states; each with a five-level response scale. The questions and responses are summarized in Table 1 below:

Scores of the 10 items are then summed, yielding a minimum score of 10 and a maximum score of 50, if respondents answer all questions. Low scores indicate low levels of psychological distress and high scores indicate high levels of psychological distress. The 2001 Victorian Population Health Survey adopted a set of cut-off scores that may be used as a guide for screening for psychological distress. Individuals with scores lower than 19 are classified as having no

**Table 1. Questions about women's emotional state, GSEPS (2009, 2014).**

|  | **About how often do you feel:** | **Response options:** |
|---|---|---|
| 1. | Tired out for no good reason? | 1. None of the time |
| 2. | Nervous? | 2. A little of the time |
| 3. | So nervous that nothing could calm you down? | 3. Some of the time |
| 4. | Hopeless? | 4. Most of the time |
| 5. | Restless or fidgety? | 5. All of the time |
| 6. | So restless you could not sit still? | |
| 7. | Depressed? | |
| 8. | Everything is an effort? | |
| 9. | So sad that nothing could cheer you up? | |
| 10. | Worthless? | |

stress; scores between 19 and 24 are indicative of mild stress; scores between 25 and 29 are indicative of moderate stress; and scores between 30 and 50 are indicative of severe stress. This cut-off classification has been adopted in studies on African countries [31, 32] and on Ghana, in particular [33, 34]. Absolute mental health scores and dummy variables for each of the four (4) mental health classifications, based on the indicated cut-offs, are employed as dependent variables in the analyses, using appropriate empirical specifications.

The GSEPS is also ideal because time-use information was collected in both waves of the survey exercise. In this study, we define domestic work as the total time spent engaging in a series of related activities within a 24-hour period on a typical working day. Women in our sample answered questions on 11 domestic activities that they performed on a typical working day. This included the amount of time spent collecting firewood, fetching water, going to the market, running other errands, doing the laundry, cleaning, cooking, taking care of elders, taking care of the sick, doing the dishes and taking primary care of children. The relevant survey question is:

> "On a typical day, does [Name] spend time [in a variety of activities. E.g. cooking, cleaning and other activities around the house]? If so, how much time does [Name] spend doing this activity (in hours and minutes)?"

The data contains other variables that are important for the present research focus such as women's demographic and socioeconomic factors, household characteristics and geographical locations. Information on these will be provided in Table 3 in Section III below.

## b. Qualitative data

The qualitative study employed both purposive and snowball sampling techniques for the selection of its participants. Participants were obtained from the Ashanti, Northern, Volta, Greater-Accra and Upper East regions. The sample selection was intended to reflect the five major ethnic groupings in Ghana, based on the country's 2010 population and housing census which considers Akans, Mole-Dagbani, Ewe, Ga-Dangme and Gurma as the five major ethnic groupings in Ghana. Participants from these ethnic groups were therefore sampled from across the regions where they are more likely to be concentrated. There are noticeable differences in socio-cultural norms and economic practices among women from the northern (i.e. Northern and Upper East regions) and southern (i.e. Greater Accra, Ashanti, Volta regions) parts of the country. Traditional gender roles tend to be more pronounced in the northern, compared to southern regions, and women tend to carry out a larger proportion, if not all, of domestic

chores. With respect to economic practices, in northern Ghana, poverty is more pronounced and gender inequality is more pronounced. This is typically demonstrated by higher incidences of early marriage, lower female enrollment rates and lower labour force participation [35]

The sampling strategy used provides the opportunity to capture the diversity in the culture and norms concerning attitudes and perceptions regarding the distribution and dynamism of household responsibilities and how it relates to women's empowerment and other related outcomes. The study relied on in-depth life-history interviews and key-informant interviews to obtain the required information.

In each region, ten (10) couples and two (2) key informants- male and female- were interviewed. In total, 110 respondents (50 couples and 10 key informants) made up the sample size. To capture heterogeneity in responses, we considered couples from low-, middle- and high-income brackets. In each region, five of the life-history interviews were conducted in urban areas while the other five were held in rural areas. All couple interviews were conducted separately although the interviews for both spouses were conducted simultaneously. The study relied on semi-structured interviews to elicit information from participants. In those localities where the researchers did not speak the local language, interviews were conducted with the assistance of interpreters. All interviews were carried out in the homes of the participants and were audio recorded with the permission of the participants.

Audio-recorded interviews were transcribed and translated verbatim into English from the local languages. For interviews conducted in English, audio recorded data were transcribed [36, 37]. Analytic rigour was ensured through comparisons of notes and recordings taken during the sessions with respondents. Thematic analysis was used to organize and categorize the data according to patterns and structures that connected the themes [36]. Thematic analysis was performed by identifying, analysing and reporting themes across the narratives. After a careful examination of the data, codes were generated and themes were developed from the text [36, 37]. An inductive approach was therefore adopted in the development of the qualitative codes where research findings were allowed to emerge from the frequent or dominant themes contained in the data. To ensure accuracy, initial themes were checked by other researchers on the research team.

The researchers carefully considered all the ethical issues involved in conducting research. Clearance for the study was obtained after going through ethical review by the Ethics Committee for the Humanities, University of Ghana. Consent was sought from all participants before proceeding with the study. Anonymity of participants are ensured by using pseudonyms to identify each respondent. Socioeconomic characteristics of study participants are summarized in Table 2 below.

As expected, the average age of the female participants is lower than the average age of their spouses. In terms of the educational attainment, close to half of all male partners in the sample have a higher level of education than the female participants. About 28% of the couples have attained the same level of education while in about 19% of couples, the female participants are more educated than their husbands or partners. With respect to earnings, close to half (44.5%) of the female participants reported that their partners earned a higher income than they did. For about 28% of the couples, the couple agreed that the women earned a higher income. There were a few instances where there was a lack of agreement as to who earned a higher income.

The data was decomposed to indicate the sectors of employment for the female participants and their spouses. About two-thirds (63.8%) of the couples both work in the informal sector. Only ten out of the forty-seven couples (21.3%) both work in the formal sector. The female participants are engaged in the informal sector as traders, food vendors and hairdressers or

**Table 2. Characteristics of participants.**

| Characteristics of Participants | Frequency (%) |
|---|---|
| **Age**: | |
| Woman Older than husband/partner | 4 (8.5%) |
| Man Older than wife/partner | 38 (80.9%) |
| Couple is the same age | 2 (4.3%) |
| Don't Know | 3 (6.4%) |
| **Total** | **47** |
| **Education** | |
| Woman more educated than husband/partner | 9 (19.1%) |
| Man has higher education than wife/partner | 25 (53.2% |
| Couple have same education level | 13 (27.7%) |
| Total | **47** |
| **Earnings** | |
| Woman earns more than husband/partner | 13 (27.7%) |
| Man earns more than wife/partner | 21 (44.5%) |
| Both say the partner earns more | 4 (8.5%) |
| Both say the partner earns less | 3 (6.4%) |
| Don't Know | 2 (4.3%) |
| Conflict in response | 4 (8.5%) |
| **Total** | **47** |
| **Employment Type** | |
| Both in formal employment | 10 (21.3%) |
| Both in informal employment | 30 (63.8%) |
| Woman in formal employment and spouse is in informal employment | 3(6.4%) |
| Woman in informal and spouse in formal | 4 (8.5%) |
| **Total** | **47** |
| **Living arrangement** | |
| Own house | 15 (31.9%) |
| Rented house/ Apartment | 6 (12.8%) |
| Compound family house | 26 (55.3%) |
| **Total** | **47** |
| **Religion** | |
| Christian | 33 (70.2%) |
| Islam | 12(25.5%) |
| Traditional (conflict) | 2 (4.3%) |
| **Total** | **47** |
| **Number of Children** | |
| Maximum | 8 |
| Minimum | 1 |
| Average | 3.4 |
| **Total** | **47** |

seamstresses, while the male participants are involved in the informal sector mainly as artisans, farmers and businesspeople. In only three instances do the female participants work in the formal sector while their partners work in the informal sector. Similarly, in four instances, the women worked in the informal sector while their partners work in the formal sector.

Between married couples, in the general Ghanaian culture, women are primarily responsible for carrying out domestic work. Such conventions can be more strongly adhered to

**Table 3. Summary statistics by year and domestic work.**

| | Full Sample | | 2009 | | 2014 | | Low Housework | | Medium Housework | | High Housework | |
|---|---|---|---|---|---|---|---|---|---|---|---|---|
| | Mean | SD | Mean | SD | Mean | SD | Mean | SD | Mean | SD | Mean | SD |
| Dependent Variables: | | | | | | | | | | | | |
| Mental health scores | 16.261 | 5.41 | 16.258 | 5.26 | 16.264 | 5.53 | 15.392 | 5.17 | 16.359 | 5.37 | 17.167 | 5.58 |
| No stress | 0.752 | 0.43 | 0.763 | 0.43 | 0.743 | 0.44 | 0.795 | 0.4 | 0.754 | 0.43 | 0.698 | 0.46 |
| Mild stress | 0.163 | 0.37 | 0.159 | 0.37 | 0.166 | 0.37 | 0.145 | 0.35 | 0.161 | 0.37 | 0.185 | 0.39 |
| Moderate stress | 0.063 | 0.24 | 0.056 | 0.23 | 0.069 | 0.25 | 0.043 | 0.2 | 0.062 | 0.24 | 0.088 | 0.28 |
| Severe stress | 0.022 | 0.15 | 0.022 | 0.15 | 0.022 | 0.15 | 0.016 | 0.13 | 0.023 | 0.15 | 0.029 | 0.17 |
| Main Independent Variable: | | | | | | | | | | | | |
| Domestic work (minutes) | 409.659 | 266.84 | 454.679 | 260.74 | 371.311 | 266.02 | - | - | - | - | - | - |
| Other Control Variables: | | | | | | | | | | | | |
| Poor household | 0.203 | 0.4 | 0.242 | 0.43 | 0.171 | 0.38 | 0.193 | 0.4 | 0.204 | 0.4 | 0.215 | 0.41 |
| Woman age | 34.542 | 15.99 | 33.293 | 15.08 | 35.606 | 16.66 | 35.448 | 18.49 | 33.748 | 15.03 | 34.472 | 13.78 |
| Never married | 0.346 | 0.48 | 0.341 | 0.47 | 0.35 | 0.48 | 0.417 | 0.49 | 0.346 | 0.48 | 0.263 | 0.44 |
| Currently married | 0.479 | 0.5 | 0.506 | 0.5 | 0.455 | 0.5 | 0.376 | 0.48 | 0.488 | 0.5 | 0.589 | 0.49 |
| Previously married | 0.175 | 0.38 | 0.152 | 0.36 | 0.194 | 0.4 | 0.207 | 0.41 | 0.166 | 0.37 | 0.148 | 0.35 |
| No education | 0.016 | 0.13 | 0.008 | 0.09 | 0.024 | 0.15 | 0.019 | 0.14 | 0.014 | 0.12 | 0.016 | 0.13 |
| Primary education | 0.313 | 0.46 | 0.303 | 0.46 | 0.322 | 0.47 | 0.312 | 0.46 | 0.296 | 0.46 | 0.337 | 0.47 |
| Secondary education | 0.63 | 0.48 | 0.65 | 0.48 | 0.614 | 0.49 | 0.623 | 0.48 | 0.652 | 0.48 | 0.612 | 0.49 |
| Post-secondary education | 0.04 | 0.2 | 0.039 | 0.19 | 0.041 | 0.2 | 0.046 | 0.21 | 0.039 | 0.19 | 0.034 | 0.18 |
| Urban residence | 0.469 | 0.5 | 0.456 | 0.5 | 0.481 | 0.5 | 0.545 | 0.5 | 0.494 | 0.5 | 0.349 | 0.48 |
| Household size | 4.075 | 1.98 | 4.253 | 2.02 | 3.923 | 1.94 | 4.003 | 2.04 | 3.979 | 1.87 | 4.281 | 2.03 |
| No physical disability | 0.732 | 0.44 | 0.69 | 0.46 | 0.768 | 0.42 | 0.785 | 0.41 | 0.746 | 0.44 | 0.653 | 0.48 |
| Employed | 0.099 | 0.3 | 0.11 | 0.31 | 0.089 | 0.29 | 0.107 | 0.31 | 0.098 | 0.3 | 0.091 | 0.29 |
| Christian | 0.878 | 0.33 | 0.876 | 0.33 | 0.88 | 0.33 | 0.883 | 0.32 | 0.883 | 0.32 | 0.866 | 0.34 |
| Muslim | 0.091 | 0.29 | 0.093 | 0.29 | 0.089 | 0.29 | 0.093 | 0.29 | 0.087 | 0.28 | 0.094 | 0.29 |
| Traditional/ None | 0.031 | 0.17 | 0.031 | 0.17 | 0.031 | 0.17 | 0.024 | 0.15 | 0.03 | 0.17 | 0.04 | 0.2 |
| Akan | 0.591 | 0.49 | 0.609 | 0.49 | 0.576 | 0.49 | 0.626 | 0.48 | 0.611 | 0.49 | 0.524 | 0.5 |
| Ga | 0.096 | 0.29 | 0.091 | 0.29 | 0.1 | 0.3 | 0.087 | 0.28 | 0.093 | 0.29 | 0.11 | 0.31 |
| Ewe | 0.128 | 0.33 | 0.111 | 0.31 | 0.143 | 0.35 | 0.092 | 0.29 | 0.128 | 0.33 | 0.17 | 0.38 |
| Northern ethnicity | 0.185 | 0.39 | 0.19 | 0.39 | 0.181 | 0.39 | 0.195 | 0.4 | 0.168 | 0.37 | 0.195 | 0.4 |
| Western | 0.111 | 0.31 | 0.131 | 0.34 | 0.095 | 0.29 | 0.053 | 0.22 | 0.125 | 0.33 | 0.163 | 0.37 |
| Central | 0.096 | 0.29 | 0.094 | 0.29 | 0.097 | 0.3 | 0.061 | 0.24 | 0.101 | 0.3 | 0.129 | 0.34 |
| Greater Accra | 0.139 | 0.35 | 0.146 | 0.35 | 0.132 | 0.34 | 0.145 | 0.35 | 0.141 | 0.35 | 0.129 | 0.34 |
| Volta | 0.078 | 0.27 | 0.062 | 0.24 | 0.091 | 0.29 | 0.047 | 0.21 | 0.088 | 0.28 | 0.101 | 0.3 |
| Eastern | 0.138 | 0.34 | 0.122 | 0.33 | 0.151 | 0.36 | 0.142 | 0.35 | 0.132 | 0.34 | 0.141 | 0.35 |
| Ashanti | 0.207 | 0.41 | 0.213 | 0.41 | 0.202 | 0.4 | 0.273 | 0.45 | 0.227 | 0.42 | 0.105 | 0.31 |
| Brong Ahafo | 0.114 | 0.32 | 0.112 | 0.31 | 0.116 | 0.32 | 0.173 | 0.38 | 0.084 | 0.28 | 0.081 | 0.27 |
| Northern | 0.05 | 0.22 | 0.05 | 0.22 | 0.051 | 0.22 | 0.035 | 0.18 | 0.04 | 0.2 | 0.081 | 0.27 |
| Upper East | 0.047 | 0.21 | 0.047 | 0.21 | 0.046 | 0.21 | 0.057 | 0.23 | 0.038 | 0.19 | 0.046 | 0.21 |
| Upper West | 0.021 | 0.14 | 0.023 | 0.15 | 0.019 | 0.14 | 0.015 | 0.12 | 0.024 | 0.15 | 0.022 | 0.15 |
| **Observations** | **5298** | | **2437** | | **2861** | | **1820** | | **1941** | | **1537** | |

especially when the couple live with other members of (the husbands) family. Therefore, the living arrangement of the couple may influence how much domestic work couples do. Over a half (55.3%) of the couples interviewed live in a compound house or a family house. The remaining couples either live in their own house (31.9%) or a rented house (12.8%).

In the sample, couples appear to practice the same religion- Christian women have Christian spouses. Similarly, Muslim women have partners who are also Muslim. In the Upper East Region, however, there are two instances where Christian women have partners who identify with traditional religion. The majority (about 70.2%) of couples identified as Christians while ten out of the forty-one couples (25.5%) are Muslim. Most of the Muslim participants are found in the northern region of the country as expected, based on the housing and population census. Reflective of the total fertility rate in Ghana, for most couples, the average number of children was three although some couples (particularly those in the Northern Region) have as many as eight children.

## c. Methods

The paper examines the effects of time spent in domestic work by women on their mental health outcomes. Mental health is represented in two ways: 1) A continuous variable- mental health scores derived from the Kessler Psychological Distress Scale (K10) discussed above; and 2) Dummy variables of mental health conditions, which are mutually exclusive, based on the 2001 Victorian Population Health Survey.

In the first instance, given the count nature of the mental health scale (i.e. ranges from 0 to 50), we employ fixed effects Poisson estimations. The model is specified as follows:

$$\text{MHealth}_{it} = \alpha_1 \text{Time}_H \text{ousework}_{it} + \alpha_2 X_{it} + \alpha_3 Z + \varepsilon_{it}$$

Where $\text{MHealth}_{it}$ refers to mental health scores for the ith woman at time t. $X_{it}$ refers to time-variant controls included in the model; Z refers to observed, time-invariant variables. $\varepsilon_{it}$ represents the error term.

In the second instance where the dependent variables are dummy variables and mutually exclusive, a multinomial logistic regression model, with fixed effects, is used. The benefit of a fixed effects specification is the ability to control for the effects of time-invariant variables with time-invariant effects. Panel regressions are more advantageous as they help to correct for potentially omitted variables in the model (e.g. previous/ family history of mental health). We assume, however, that these variables do not change over time and have the same effect over time.

Given that women with poorer mental health may be less able to spend time in domestic work, a most important factor to address is the potential self- selection of healthier women into household work. We therefore approximate causal inference through the use of a quasi-experimental technique- propensity score matching. We consider women who spend more time in "arduous" domestic work as belonging to a "treatment group" while other women who spend less time doing domestic work are assigned to the "comparison/control group". K10 mental health scores are arranged in order from lowest to highest; the data is then partitioned into three equal groups or terciles. Women in the highest tercile are classified as performing arduous domestic work and assigned to the treatment; women in lower terciles are assigned to the control group. In an alternative specification, we assign women in the lowest domestic work tercile to the treatment group and women in higher terciles to the comparison group. We expect that controlling for selection, women in the highest (lowest) work tercile will be associated with higher (lower) mental distress.

Propensity scores are generated from observable characteristics of women, which are then used to produce a pseudo "control" group of women that are similar to the treatment group. This way, the selection of women into the treatment group more closely resembles a random assignment. We will use these propensity scores, combined with appropriate matching techniques to further reduce selection bias and estimate the average treatment of arduous work on

the treated (i.e. ATT). Assume $MH_{i,T}$ is the mental health status of woman i in the treatment group and $MH_{i,C}$ is the mental health status for a woman in the control group. The difference in mental health may be given as $MH_{i,T}-MH_{i,C}$. We are interested in the average effect of the treatment for those who received the treatment (i.e. $E[MH_{i,T}-MH_{i,C} \mid Ti = 1]$. However, this is impossible to estimate, given the difficulty in observing the counterfactuals (i.e. $MH_{i,T}$ for $Tt = 0$ and $MH_{i,C}$ for $Tt = 1$ are not observed). To address this problem and generate appropriate counterfactuals, we assume that conditional on observed characteristics $Xi$, women's time spent in housework is independent of their mental health status. Propensity scores are then constructed using a probit model and used to match treatment units with observationally similar control units (i.e. $P(Xt) = Pr(Tt = 1 \mid Xi)$). Balancing on the propensity scores will eliminate selection bias based on the observable characteristics, $Xi$. It is important to emphasize, as a caveat, that matching cannot be carried on women's unobservable characteristics, however.

## III. Results

This section is made up of three parts. The first part presents summary statistics of study variables for the analytical sample, disaggregated by the survey year and domestic work terciles. We also show results for both fixed effects and multinomial logistic regressions in the next part. Finally, as a robustness check, we provide estimates from the propensity score matching technique, which accounts for potential self-selection in the relationship between women's health and their carrying out of domestic responsibilities.

### a. Descriptive results

Summary statistics of study variables for the analytic sample are provided in Table 3 for each survey year and also by domestic work terciles. The average mental health score for women in the sample is 16 and falls in the range of little or no stress. The higher the score, the poorer a woman's mental health. Seventy-five percent of women fall in the range of "little or no stress". Sixteen percent of women fall in the "mild stress" range, while 6% and 2% of women fall in the "moderate" and "severe" stress categories, respectively. The main independent variable of interest is the amount of time that women spend in domestic activities. Although women spend about 7 hours a day on domestic work, including primary childcare, this duration has been falling over time. Furthermore, mental health scores are lower with lower domestic work terciles.

Other controls are included in the analyses: Using data on household expenditures and the national poverty line, it is observed that 20% of women in the sample belong to poor households, with the proportion of poor households declining between 2009 and 2014. Poverty appears to be correlated with more time spent in housework. On average, women in the sample are about 35 years of age and almost half the sample is married. It appears that women do more housework when they are currently married, on average, compared to when they are not.

With respect to education, majority of women have primary (31%) and secondary (63%) education; almost half of the sample resides in urban areas with an average household size of 4 members. Close to 10% of women are in paid employment and almost a quarter report that they do not have any physical disabilities that may interfere with their daily activities. Controls are also included for religion, ethnicity and regions of residence.

### b. Results from regressions and qualitative data

Regression results indicate that an extra minute spent in domestic work is associated with worse mental health outcomes among women in Ghana (see Table 4). This has been linked to routinization, depreciation and constant interruptions of tasks [7].

**Table 4. Fixed effects of regression of mental health scores, with and without SES interactions.**

| | Dependent variable is mental health scores: | |
|---|---|---|
| | **Without SES Interactions** | **With SES Interactions** |
| Domestic work | 0.0001*** | -0.0001* |
| | (4.97) | (-1.81) |
| High SES | -0.0013 | -0.1257*** |
| | (-0.07) | (-3.90) |
| Domestic work*High SES | - | 0.0003*** |
| | | (4.76) |
| Woman age | 0.0012 | -0.0002 |
| | (0.20) | (-0.04) |
| Woman age (squared) | -0.0000 | -0.0000 |
| | (-0.43) | (-0.17) |
| Current Married | 0.0174 | 0.0190 |
| | (0.47) | (0.52) |
| Previously married | 0.0554 | 0.0564 |
| | (1.35) | (1.37) |
| Primary education | -0.1965*** | -0.1936*** |
| | (-2.88) | (-2.84) |
| Secondary education | -0.1830*** | -0.1795*** |
| | (-2.63) | (-2.58) |
| Post-secondary education | -0.1366* | -0.1244 |
| | (-1.67) | (-1.52) |
| Urban residence | 0.0131 | 0.0047 |
| | (0.22) | (0.08) |
| # of Household members | -0.0011 | -0.0031 |
| | (-0.18) | (-0.50) |
| No physical disability | -0.1818*** | -0.1868*** |
| | (-11.93) | (-12.22) |
| Paid employment | 0.0250 | 0.0282 |
| | (0.94) | (1.06) |
| Ga | 0.1779** | 0.1784** |
| | (2.40) | (2.41) |
| Ewe | -0.0238 | -0.0226 |
| | (-0.24) | (-0.22) |
| Northern tribes | 0.1867*** | 0.1989*** |
| | (2.67) | (2.84) |
| Year (2014) | 0.0073** | 0.0072** |
| | (2.35) | (2.31) |
| Religion controls | YES | YES |
| R-squared (within) | 0.1040 | 0.1153 |
| R-squared (between) | 0.0809 | 0.0835 |
| R-squared (overall) | 0.0787 | 0.0823 |
| Number of observations | 5,298 | 5,298 |

T-statistics in parenthesis

* $p < 0.10$

** $p < 0.05$

*** $p < 0.01$.

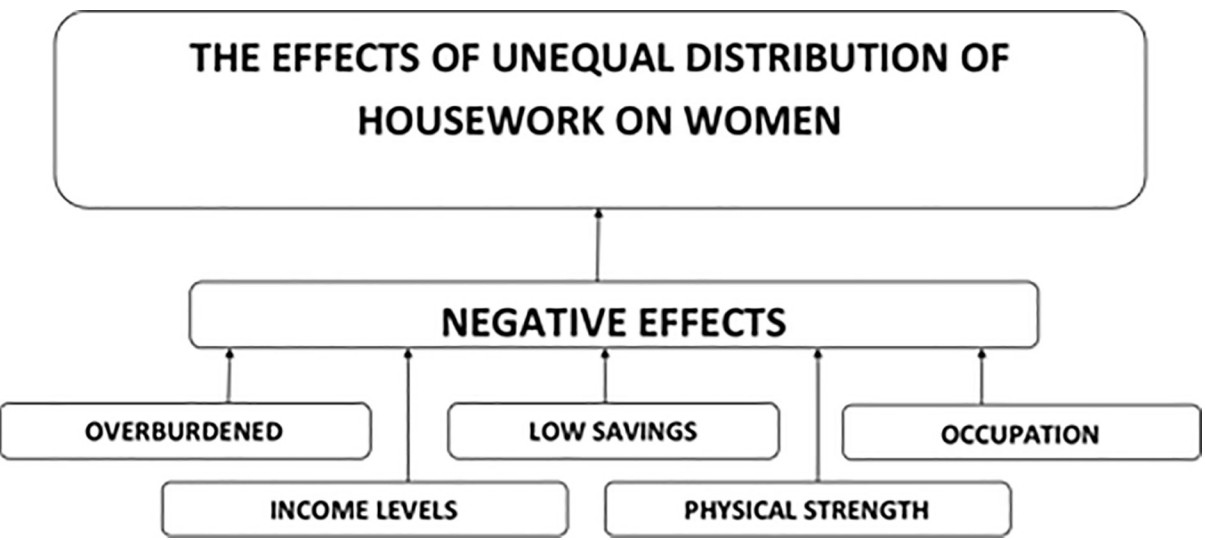

**Fig 2. Themes arising from qualitative survey of women about negative effects of unequal housework.**

This finding is complemented by findings from the qualitative survey of married/cohabiting women in Ghana. Analyses of qualitative surveys identified various themes relating to the negative effects of unequal distribution of domestic work and childcare burdens on women. These included feelings of being overburdened; physical exhaustion, in addition to others like limited participation in labour markets, leading to loss of income (see Fig 2 below).

**i. Overburdened.** The majority of women interviewed agreed that the most important effect of unequal domestic and childcare responsibilities was the considerable stress that activities placed on them.

"... It was great suffering for me; I was overburdened since I was the only taking care of everything!" (Wife from Volta region)

"... It's a lot! There's stress in it my work and my responsibility of taking care of the children coupled with my domestic chores..." (Wife from Volta region)

"... Because all that was work, you will be doing several things and it was stressful. I will say the way it was stressful it makes you fall sick..." (Wife from Greater Accra region)

"... The women are the ones suffering the most these days when it comes to working and fending for the family......" (Key Informant from Ashanti region)

With respect to childcare, women also reported being stressed from caring for younger children:

"... Yes, I felt overburdened by the fact that now aside my domestic work and then working outside for money and I have an additional care to do, that is childcare, by all means. The roles increase because aside the domestic chores, the working environment, you still have to care for the baby, changing of diapers, washing of cloth. Every blessed day you have to wash the napkins and other things. So, it was actually overburdened because sometimes you will be washing in the night, that is when you close from work and come, after cooking you have to wash all and dry them before the next day..." (Wife from Upper East region)

**ii. Physical effects.** Although not as prominent as being mentally stressed, some women also reported some effects on their physical health.

"... Because all that was work, you will be doing several things and it was stressful. I will say the way it was stressful it makes you fall sick.. .." (Wife from Greater Accra region)

"... On the part of health, I am really not healthy the work has made my body to ache so it is a burden because I am not healthy like I should be and the way marriage is, it is difficult and between us it is not good. I don't know what to say it is not anything good because we are not able to sit together to chat.. .. ..." (Wife from Upper East region)

"... Sometimes you wake up with bodily pains all over.. ..." (Wife from Volta region)

**iii. Economic effects (i.e. income levels, low savings and occupational effects).** There is an economic effect on women because they either stop working or change jobs with lower earnings as they cannot combine these with their domestic responsibilities.

"...Yes, taking care of a child, cooking, washing and performing house chores affected my income levels or business. I was not getting offers regularly because the time I am needed at the shop I am not there.. .." (Wife from Greater Accra region)

"... yes, it was really a burden for me because I had to wake up very early because if I went to work late, I would be fined. And sometimes when the day is coming to an end around 4pm, that was when there was more business but I had to close and come home to cook.. .. ..." (Wife from Ashanti region)

"... This has affected my business. I have stopped my earlier business this time I do sell oranges which is relatively easier for me.. .." (Wife from Northern region)

Our general findings appear to be consistent with the role conflict theory, where women, burdened with childcare and domestic work and combined with the demands of labour work, display more depression symptoms as they generally receive fewer role-related rewards and more role-related stresses [17].

From the results in Table 4 above, it is also observed that women with higher socioeconomic status (SES) generally have better mental health outcomes likely because they are able to hire domestic workers and/or purchase domestic technology to reduce time spent in these domestic and childcare responsibilities. This presumption is supported by women's responses from the qualitative interviews that were conducted.

"...Like when I am cooking, instead of wasting time to start a fire I use the gas stove. Instead of grinding, you can blend, and it is fast. ...so it has helped me a lot.. .." (Wife from rural Greater Accra)

"...Yes, because I can pay, I employ the domestic helps. It relieves me.. .. ..." (Wife from Upper East region)

"...Yes, I had blender and rice cooker, I had mine. Yes, it helped me a lot. It makes things faster ..." (Wife from Ashanti region)

In Table 5, we run a multinomial logistic regression model with fixed effects and report odds ratios with robust errors. Results indicate that a one-minute increase in the time spent by

**Table 5. Multinomial logistic regressions with fixed effects, with and without SES interactions.**

| | Base group is "No stress" | |
| --- | --- | --- |
| | **Without interactions** | **With interactions** |
| **Mild Stress** | | |
| Domestic work | 1.0009*** | 1.0004 |
| | (2.92) | (0.68) |
| High SES | 1.6878** | 1.2960 |
| | (2.42) | (0.74) |
| Domestic work*High SES | - | 1.0006 |
| | | (0.89) |
| Woman age | 1.0782*** | 1.0761*** |
| | (2.72) | (2.64) |
| Woman age (squared) | 0.9993** | 0.9993** |
| | (-2.19) | (-2.12) |
| Currently Married | 1.0355 | 1.0599 |
| | (0.13) | (0.21) |
| Previously married | 1.2969 | 1.2916 |
| | (0.83) | (0.82) |
| Primary education | 0.8265 | 0.8329 |
| | (-0.33) | (-0.31) |
| Secondary education | 0.8162 | 0.8332 |
| | (-0.35) | (-0.32) |
| Post-secondary education | 0.5113 | 0.5231 |
| | (-0.94) | (-0.90) |
| Urban residence | 0.8124 | 0.7981 |
| | (-0.33) | (-0.36) |
| Number of household members | 0.9210 | 0.9184 |
| | (-1.26) | (-1.30) |
| No disability | 0.3466*** | 0.3432*** |
| | (-6.84) | (-6.85) |
| Employed | 0.6836 | 0.6916 |
| | (-1.43) | (-1.39) |
| **Moderate Stress** | | |
| Domestic work | 1.0016*** | 0.9992 |
| | (2.98) | (-0.72) |
| High SES | 1.5531 | 0.4148 |
| | (1.23) | (-1.37) |
| Domestic work*High SES | - | 1.0031** |
| | | (2.43) |
| Woman age | 1.0402 | 1.0253 |
| | (0.82) | (0.52) |
| Woman age (squared) | 0.9996 | 0.9998 |
| | (-0.68) | (-0.44) |
| Currently Married | 1.9889 | 2.2747* |
| | (1.44) | (1.71) |
| Previously married | 2.4229* | 2.8094* |
| | (1.68) | (1.95) |
| Primary education | 1.5290 | 1.6919 |
| | (0.45) | (0.54) |

(*Continued*)

**Table 5.** (Continued)

| | Base group is "No stress" | |
|---|---|---|
| | **Without interactions** | **With interactions** |
| Secondary education | 1.4485 | 1.6344 |
| | (0.39) | (0.50) |
| Post-secondary education | 1.9983 | 2.8058 |
| | (0.57) | (0.83) |
| Urban residence | 0.6186 | 0.5003 |
| | (-0.38) | (-0.54) |
| Number of household members | 1.0212 | 1.0032 |
| | (0.20) | (0.03) |
| No disability | 0.2994*** | 0.2824*** |
| | (-4.88) | (-4.94) |
| Employed | 0.6874 | 0.7127 |
| | (-0.85) | (-0.75) |
| **Severe Stress** | | |
| Domestic work | 1.0002 | 0.9981 |
| | (0.17) | (-1.20) |
| High SES | 0.8719 | 0.2462 |
| | (-0.23) | (-1.42) |
| Domestic work*High SES | - | 1.0031* |
| | | (1.69) |
| Woman age | 0.9638 | 0.9540 |
| | (-0.40) | (-0.51) |
| Woman age (squared) | 1.0006 | 1.0007 |
| | (0.54) | (0.62) |
| Currently Married | 1.3076 | 1.3638 |
| | (0.28) | (0.32) |
| Previously married | 7.2579* | 8.2494* |
| | (1.76) | (1.84) |
| Primary education | 0.0591* | 0.0412** |
| | (-1.86) | (-2.11) |
| Secondary education | 0.0948 | 0.0790 |
| | (-1.46) | (-1.59) |
| Post-secondary education | 0.0000 | 0.0000 |
| | (-0.01) | (-0.01) |
| Urban residence | 0.9607 | 0.9800 |
| | (.) | (.) |
| Number of household members | 1.7930** | 1.7671** |
| | (2.34) | (2.27) |
| No disability | 0.3172** | 0.2912** |
| | (-2.40) | (-2.50) |
| Employed | 0.0000 | 0.0000 |
| | (-0.01) | (-0.01) |
| Religious Controls | YES | YES |
| Ethnicity Controls | YES | YES |
| Wave Controls | YES | YES |

(*Continued*)

**Table 5.** (Continued)

| | Base group is "No stress" | |
| --- | --- | --- |
| | **Without interactions** | **With interactions** |
| Number of Observations | 1,728 | 1,728 |

T-statistics in parenthesis

* p<0.10

** p<0.05

*** p<0.01.

women in domestic work, increases the odds of mild and moderate stress, versus no stress, consistent with results from Table 4 above.

The inclusion of interaction effects between women's SES status and domestic work, again, provides interesting effects on their mental health. In Table 5, as was observed in Table 4, women with higher SES status who spend more time in domestic work have higher odds of stress. In Table 5, this increases the odds of moderate and severe stress, although the results are only marginally significant in the latter specification.

The finding that women of higher socioeconomic class with increasing time spent in domestic work had higher odds of depression initially appears counterintuitive- women in wealthier homes should be able to easier afford house helps and domestic technology, both of which would reduce time spent in housework and subsequently, stress levels. Findings how-ever remain plausible for a number of reasons. First, women's mental distress may stem from equity-based considerations. Women from wealthier households are more empowered [38]. Existing research suggests that men from wealthier households are less likely to help out with domestic work [21]. Given that fairness in housework has been identified as a critical determi-nant of depression in wives [5], wealthier women may be more distressed from doing house-work if the division of housework is perceived as being unfair and inegalitarian. Indeed, [39] show that husbands' involvement in housework is negatively associated with wives' psycholog-ical distress.

A second explanation is the opportunity cost of domestic work among wealthier women. Wealthier women may be engaged in work which brings in significant earnings; time spent in housework may therefore have higher costs and lead to more stress. Alternatively, wealthier women may prefer to spend their time in more leisurely activities like relaxing, shopping, among others. Time spent in housework may increase stress levels if they cut into these periods of leisure and relaxation. A third explanation may be that women from wealthier households may not have had much experience with housework, to begin in, given a possibly more indulged upbringing. If this is the case, time spent in housework, including childcare, is likely to cause some associated discomfort and distress from their involvement. Finally, wealthier women may be more involved in labour market work, the combination with domestic work may therefore overburden her and lead to more mental distress.

Results from the qualitative survey however provide a different perspective from the fourth explanation given above. Interviews with key informants describe a situation where women who are actively engaged in the labour market, and may have relatively higher SES status, actu-ally have less time for domestic chores and may therefore suffer less mental stress from bur-dens of housework.

". . .when she wakes up in the morning, she is in a hurry to go and make money and has less time for chores. . . . . .." (Key Informant from Rural Greater Accra)

This observation by the key informant therefore suggests that not all women feel bound by social and cultural expectations to carry out domestic work. In some cases, the need for economic sustenance may overshadow the need to carry out domestic responsibilities and reduce women's time spent in these activities.

Other results are worth noting- women initially have higher odds of mental health up to a certain age, and then the odds decrease thereafter. According to the WHO, depression and poor mental health is almost always reported to be twice as common in women as in men across diverse societies and social contexts. Additionally, gender differences in rates of depression are strongly age related. The largest differences occur in adult life; no differences are found in childhood and few in the elderly [40–42]. This is likely because women of reproductive age may carry the triple burden of productive, reproductive and care work.

The lack of a physical disability lowers odds of poor mental health [43]. Currently married and previously married women have higher odds of moderate stress, versus no stress, compared to never married women [44]. Compared to women with no education, women with primary education had lower odds of severe stress [45]. This is because access to schooling may have a direct effect on mental and psychological health, through improvements in self-esteem and life choices. Finally, women who resided in households with more members were associated with higher odds of severe stress [46].

## c. Robustness checks for selection and reverse causality

Given that women with poorer mental health may be less able to spend time in domestic work, we control for self- selection of healthier women into more arduous household work as a robustness check on our results. We therefore employed a propensity score matching technique to correct for this. First, as described above, we calculated the propensity scores, based on the likelihood that women are involved in arduous work, and then matched women with similar propensity scores. Matching was done using nearest neighbours with replacement for each observation. Here, a woman from the comparison group is chosen as a matching partner for a treated woman that is closest in terms of their propensity score. As shown in Fig 3 below, treatment and control groups have been well-matched; similarities in their properties indicate that the effect of being in a treatment or control group is arbitrary.

Table 6 below shows results of the treatment effect on the treated (ATT). Results from the PSM technique show a statistically significant negative (positive) effect on mental health for the average treatment effect on the treated (ATT), indicating that women in more (less)

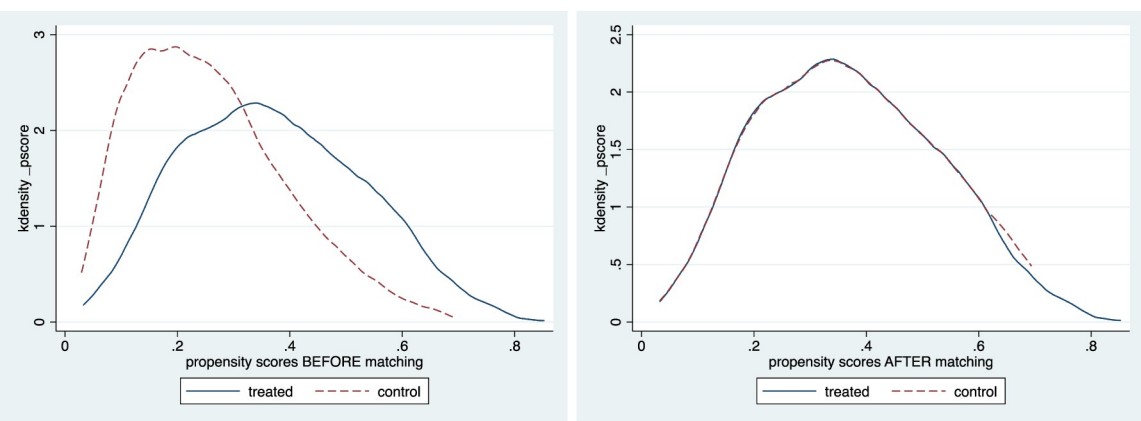

**Fig 3. Nearest neighbour matching.**

**Table 6. Propensity score estimates.**

| Mental health scores as dependent variable: | | |
| --- | --- | --- |
| Arduous housework (i.e. women in highest domestic work tercile) | Unmatched sample | 1.276*** (7.83) |
| | Matched sample | 0.535** (2.01) |
| Less arduous housework (i.e. women in lowest domestic work tercile) | Unmatched sample | -1.324*** (-8.51) |
| | Matched sample | -1.0972*** (-4.52) |
| Sample size | 5,298 | |

T-statistics in parenthesis

* p<0.10

** p<0.05

*** p<0.01.

arduous domestic work are more likely to have worse (better) mental health, compared to "similar" women engaged in less (more) arduous housework. The magnitudes of the effects are however smaller under the PSM approach, compared to the unmatched models, indicating some effects of selection bias in the latter.

## IV. Discussion of results

Results from both the qualitative and quantitative analyses indicate that women suffer adverse mental effects from domestic and childcare responsibilities. Using a propensity score matching technique, we controlled for self- selection of healthier women into more arduous household work as a robustness check on our results and found that results were robust to selectivity concerns. Contrary to expectations, we found that domestic and childcare work was particularly stressful among women from wealthier backgrounds. We reasoned that, among other reasons, this could be attributable to higher opportunity costs of domestic work involvement.

Reasons for the continued perpetration of unequal housework burdens are contained in entrenched traditional gendered roles for men and women. As indicated in Appendix 1, community expectations are a driving force behind the perpetuation of unequal domestic work-loads for men and women in Ghanaian households. Some men refuse to help their wives due to fear of mockery. In other cases, however, some male partners do provide assistance with domestic activities as a show of support to their wives. A number of reasons have been given for increased assistance by male partners over time such as the effects of education and spread of Western values.

Results also indicated that women suffer adverse effects on their physical health. This is contrary to other researchers who have argued that performing housework is synonymous with physical activity and therefore better health [6, 18]. Interviews with women in Ghana indicate that the level of housework, often combined with childcare and market work leads to significant levels of physical discomfort. We also found that in addition to adverse mental and physical effects, women also suffer reduced labour market participation and income levels as a result of their domestic work burdens. Women will often have to give up apprenticeships, reduce their involvement in the labour market, or seek employment in more flexible but low-paying jobs in an attempt to balance these activities with their housework responsibilities. This has implications for their income earning capacities and potentially, the welfare of household members.

## V. Conclusion

This study highlighted the need for greater research into the unpaid economy, particularly in the developing country context. The study finds that increased burden of domestic work has

important implications for a woman's wellbeing, even when controlling for self-selectivity. A reason for women's greater stress is often the difficulty in combining domestic work with labour market responsibilities. The qualitative study highlighted a number of important arguments- First, women in Ghana do feel overburdened and stressed from the amount of unpaid work that they engage in. Second, although this research focuses on the effect of domestic and childcare responsibilities on women's mental health, it is evident that these activities also take a toll on their physical wellbeing. Third, we also found that domestic work burdens have negative effects on women's labour force participation and their income levels.

With close to 70% of women engaged in some form of market work in Ghana, more efforts should be made to encourage greater participation by men in domestic work. Although the tide is changing and more men are recognizing the stress that women face from the double burden of labour work and domestic responsibilities, women themselves often discourage help from their partners, for fear of societal reprisals. The classification of domestic work as economic inactivity and the sole preserve of women contributes to the inevitable overload.

Gender specific roles are largely culturally prescribed and in many cases men and women themselves adhere strictly to these, even in the face of deleterious effects on women's health. It is important to redirect the socialization of men and women, particularly during formative years, to encourage male partners' assistance with domestic work, as this is shown to have positive implications for women's health.

Other findings from the research suggest that access to labour saving devices was also noted to considerably decrease women's mental distress as these simplified various tasks and shortened the time spent in carrying out domestic activities.

## Supporting information

**S1 File. Thematic network on understanding the distribution of domestic responsibilities in the Ghanaian setting and changes over time.**
(PDF)

## Author Contributions

**Conceptualization:** Nkechi S. Owoo.

**Formal analysis:** Nkechi S. Owoo, Monica P. Lambon-Quayefio.

**Methodology:** Monica P. Lambon-Quayefio.

**Writing – original draft:** Nkechi S. Owoo, Monica P. Lambon-Quayefio.

**Writing – review & editing:** Nkechi S. Owoo, Monica P. Lambon-Quayefio.

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
