## [Decision Letter · Decision Letter 0]

7 Oct 2020

PONE-D-20-22368

Mixed Methods Exploration of Women’s Housework and Effects on their Mental Health

PLOS ONE

Dear Dr. Lambon-Quayefio,

Thank you for submitting your manuscript to PLOS ONE. After careful consideration, we feel that it has merit but does not fully meet PLOS ONE’s publication criteria as it currently stands. Therefore, we invite you to submit a revised version of the manuscript that addresses the points raised during the review process.

ACADEMIC EDITOR: All three reviewers find there is a merit in your manuscript. However, they come up with several suggestions which I felt are very important. If your willing to revise and resubmit the paper according to reviewer comments, we are willing to consider this.  The manuscript fails to triangulate both quantitative and qualitative data as it claims to offer.  Please take this suggestion and revise the paper carefully. However, this doesn't mean the revised version is automatically accepted. It will be re-evaluated by me and the reviewers if needed.

We look forward to receiving your revised manuscript.

Kind regards,

Srinivas Goli, Ph.D.

Academic Editor

PLOS ONE

Journal Requirements:

2. Thank you for your ethics statement: "Name of Institutional Board: University of Ghana Ethics Committee for the Humanities (ECH)

Approval Number: ECH 127/18-19

Consent: It was written. It was read out to participants of the study and they appended signatures(or thumb print) to indicate consent."

i) Please amend your current ethics statement to confirm that your named institutional review board or ethics committee specifically approved this study.

ii) Once you have amended this/these statement(s) in the Methods section of the manuscript, please add the same text to the “Ethics Statement” field of the submission form (via “Edit Submission”).

4. Please ensure that you refer to Figure 1 in your text as, if accepted, production will need this reference to link the reader to the figure.

5. ** Please include your tables as part of your main manuscript and remove the individual files **. Please note that supplementary tables (should remain/ be uploaded) as separate "supporting information" files

Additional Editor Comments (if provided):

All three reviewers find there is merit in your manuscript. However, they come up with several suggestions which I felt are very important. If your willing to revise and resubmit the paper according to reviewer comments, we are willing to consider this. The manuscript fails to triangulate both quantitative and qualitative data as it claims to offer. Please take this suggestion and revise the paper carefully. However, this doesn't mean the revised version is automatically accepted. It will be re-evaluated by me and the reviewers if needed.

Reviewers' comments:

Reviewer's Responses to Questions

**Comments to the Author**

1. Is the manuscript technically sound, and do the data support the conclusions?

Reviewer #1: Yes

Reviewer #2: Partly

Reviewer #3: Partly

2. Has the statistical analysis been performed appropriately and rigorously? 

Reviewer #1: Yes

Reviewer #2: Yes

Reviewer #3: No

3. Have the authors made all data underlying the findings in their manuscript fully available?

Reviewer #1: No

Reviewer #2: Yes

Reviewer #3: Yes

4. Is the manuscript presented in an intelligible fashion and written in standard English?

Reviewer #1: Yes

Reviewer #2: No

Reviewer #3: No

5. Review Comments to the Author

Reviewer #1: The research paper is well written and adds to an important domain. The use of mixed methods approach, use of GESPS data and sound use of statistical tools are the strengths of the paper. However, there are some important areas which the authors may consider to look at.

1. Though the authors provided a great detail of information about GSEPS, it is important that they inform how they accessed the data, process involved and acknowledge the source of the data. It is essential since the providers of the secondary data require them to be acknowledged in the manuscripts published using the data.

2. The researchers reported the use of a total of 60 qualitative interviews (50 couples and 10 key informants). However, a limited information was provided regarding measures taken to ensure the rigor of the qualitative data collected. The researchers may please provide the detailed outline of the same. It is also advised to provide a table outlining the characteristics of the Interview participants, it will establish that participants are best fit with regard to this study.

3. The researchers need to provide a detailed outline on the data collection (if recording was used or not, if yes if consent obtained, language of the interview, validation of the interview by the respondents), and data analysis procedures followed to collect and analyze qualitative data.

Reviewer #2: I read the manuscript with great interest and found an interesting analysis of quantitative data. However, the manuscript fails to triangulate both quantitative and qualitative data as it claims to offer. I have the following main observations:

Please find attached comments

Reviewer #3: • The abstract need to be more scientific, structured with summary of key findings.

• The introduction is too extensive and superfluous. It should not be above 850 words.

• The content is not written in scientific. The objectives of this work need to be more focused into the analysis.

• Result and discussion should be following independently.

• The regression and multinomial logistic model didn’t explained proper scientific way and incomplete to proof the goals of the paper. Results are undependable as statistical results are merely stated without any analytical explanation.

• Quantitative Analysis should be more in-depth, scientific and context specific for the specific area. And study is need for more information on policy implications of these particular findings.

• It uses inappropriate statistical tools for Table 3 & 4.

6. PLOS authors have the option to publish the peer review history of their article (what does this mean?). If published, this will include your full peer review and any attached files.

Reviewer #1: No

Reviewer #2: **Yes: **Jayakant Singh

Reviewer #3: No

---

## [Author Response · Author response to Decision Letter 0]

30 Oct 2020

Working Papers Folder: Childcare Distribution and Welfare Outcomes

Journal Requirement:

Comment:

Response: The manuscript has been modified in accordance with the PLOSE ONE style as indicated in the reference documents provided 

Comment: 

Thank you for your ethics statement: "Name of Institutional Board: University of Ghana Ethics Committee for the Humanities (ECH)

Approval Number: ECH 127/18-19

Consent: It was written. It was read out to participants of the study and they appended signatures(or thumb print) to indicate consent."

i) Please amend your current ethics statement to confirm that your named institutional review board or ethics committee specifically approved this study.

ii) Once you have amended this/these statement(s) in the Methods section of the manuscript, please add the same text to the “Ethics Statement” field of the submission form (via “Edit Submission”).

Response: This has been done in the online submission portal

Comment:

We note that you have indicated that data from this study are available upon request. PLOS only allows data to be available upon request if there are legal or ethical restrictions on sharing data publicly. For information on unacceptable data access restrictions, please see http://journals.plos.org/plosone/s/data-availability#loc-unacceptable-data-access-restrictions.

 Response: Although we are happy to share the qualitative data associated with the study, the terms of the contract with the funders does not allow the researchers to make the data publicly available before the termination of the project. The data sharing arrangements per the contract with the funders is indicated as follows: “We agree to make the data publicly available as soon as possible but no later than within one year of the completion of the funded project period for the parent award or upon acceptance of the paper for publication, whichever is earlier”. Per this provision, we are unable to make the data publicly available since the project has not yet ended.

4. Please ensure that you refer to Figure 1 in your text as, if accepted, production will need this reference to link the reader to the figure.

Response:

This has been done.

Comment:

5. ** Please include your tables as part of your main manuscript and remove the individual files **. Please note that supplementary tables (should remain/ be uploaded) as separate "supporting information" files

 Response: This has been done. Thank you.

Additional Editor Comments (if provided):

All three reviewers find there is merit in your manuscript. However, they come up with several suggestions which I felt are very important. If your willing to revise and resubmit the paper according to reviewer comments, we are willing to consider this. The manuscript fails to triangulate both quantitative and qualitative data as it claims to offer. Please take this suggestion and revise the paper carefully. However, this doesn't mean the revised version is automatically accepted. It will be re-evaluated by me and the reviewers if needed.

Response:

We are grateful to the reviewers for the comments provided and have given serious considerations to each suggestion. The manuscript has been revised accordingly. We would however like to state that in more than one case, we experienced some difficulty in understanding what changes were being suggested by Reviewer 3. 

Reviewer 1

Comment:

The research paper is well written and adds to an important domain. The use of mixed methods approach, use of GESPS data and sound use of statistical tools are the strengths of the paper. However, there are some important areas which the authors may consider to look at.

Response:

Thank you.

Comment:

Though the authors provided a great detail of information about GSEPS, it is important that they inform how they accessed the data, process involved and acknowledge the source of the data. It is essential since the providers of the secondary data require them to be acknowledged in the manuscripts published using the data.

Response:

The following information is provided in the section describing the Quantitative data (i.e. Section IIa):

The Ghana Socioeconomic Panel Survey is a joint effort between the Economic Growth Centre (EGC) at Yale University and the Institute of Statistical, Social and Economic Research (ISSER), at the University of Ghana (Legon, Ghana). The survey is principally funded by the EGC, designed by both the EGC and ISSER, and carried out and supervised by ISSER. Technical support is provided by the University of Michigan’s Survey Research Center International Unit (SRC IU). Although wave 1 is publicly available, (https://microdata.worldbank.org/index.php/catalog/2534), Wave 2 data is available only upon request. Interested researchers are asked to provide information about themselves and the use to which they will put the data before access is granted.

Comment:

The researchers reported the use of a total of 60 qualitative interviews (50 couples and 10 key informants). However, a limited information was provided regarding measures taken to ensure the rigor of the qualitative data collected. The researchers may please provide the detailed outline of the same. It is also advised to provide a table outlining the characteristics of the Interview participants, it will establish that participants are best fit with regard to this study.

Response: 

Additional information has been provided on measures taken to ensure the rigor of the qualitative data collected. Additionally, a table, with some discussion, has been provided that describes the characteristics of the interviewed couples.

Comment:

The researchers need to provide a detailed outline on the data collection (if recording was used or not, if yes if consent obtained, language of the interview, validation of the interview by the respondents), and data analysis procedures followed to collect and analyze qualitative data.

Response:

This has been provided in Section IIb.

Reviewer 2

Comment:

I read the manuscript with great interest and found an interesting analysis of quantitative data. However, the manuscript fails to triangulate both quantitative and qualitative data as it claims to offer. I have the following main observations:

Comment:

The title needs to be reframed keeping in mind childcare and Ghana 

Response: 

The current title is Mixed Methods Exploration of Women’s Housework and Effects on their Mental Health. Following the reviewer’s suggestion, we are happy to amend this to the following title: Mixed Methods Exploration of Ghanaian Women’s Domestic Work, Childcare and Effects on their Mental Health

Comment:

The Introduction fails to highlight the rationale of the study. The need to do this study seems to suggest that because regression estimates would be potentially biased in the past studies and hence this paper offers solution by employing PSM approach. I am not convinced that this study is not adding any value than an analytical refinement. 

Response:

Many thanks for this observation. I have made the motivating arguments on potential mental health implications of domestic burden more explicit, particularly in the present context.

Comment:

The authors need to be careful about the use of cut-off classification for mental health. Justification of the cutoff needs to be substantiated with results of other studies in African settings. 

Response:

Many thanks for this comment. We have provided references to indicate that these cut-offs have been adopted in recent studies on both Africa in general, an on Ghana, in particular.

Comment:

Considering the study sampled participants across the five regions, cultural aspects, gender norms, gender roles etc. in these settings need to be discussed in greater detail in light of Women’s Housework and Effects on their Mental Health. Moreover, what is the regional variations and how these regions are different in socio-cultural and economic practices are missing. The study lacks adequate rational for the choice of methods used to report the study and considerably misses the social-cultural practices across different ethnic groups concerning gender roles, societal view on participation of women in labour force including the current scope and trend of women’s employment outside the household. 

Response:

A discussion on differences in sociocultural norms and economic practices between women from northern and southern Ghana has been included.

Comment:

The results need to be presented as per the emerging themes in the qualitative part of the study. While the quantitative method is elaborated somewhat descriptively but there is no adequate information about qualitative data collection process and analysis including triangulation of both data that fails to justify the design used to study the research problem. 

Response:

Thank you for this comment. We have discussed the themes more fully in the revised document. We have also provided greater information about the data collection process and triangulated both quantitative and qualitative data more effectively.

Comment:

The qualitative data is presented without any background and discussion after the quantitative data is reported. Ideally, while triangulating, both findings (quantitative and qualitative) must flow seamlessly. However, this study reports findings from quantitative and qualitative separately that fails to address the purpose of mixed method research. 

Response:

Thank you for these helpful comments. We have attempted to integrate the findings from the qualitative surveys more comprehensively into quantitative results, rather than reporting results from each analysis separately.

Comments:

The findings need to be discussed in greater details which seems completely a miss now. The inference drawn in conclusion, “more efforts should be made to lessen the incompatibilities between these dual roles. This can be done through institutional factors like the provision of childcare services.” seem out of context. Similar inferences must not be drawn.

Response:

These concerns are well-noted. We have limited our recommendations to more direct findings from the research. 

Reviewer 3

Comment:

The abstract need to be more scientific, structured with summary of key findings.

Response:

Thank you. We are however not aware that this journal requires/requests abstracts to be submitted in a structured format. The reviewer may kindly confirm via the following link: https://journals.plos.org/plosone/s/file?id=wjVg/PLOSOne_formatting_sample_main_body.pdf

Comment:

The introduction is too extensive and superfluous. It should not be above 850 words.

Response:

Thank you for your comment. The journal does not appear to have a word limit for the introduction. It is a little lengthy because in providing the motivation for the study, we have attempted to indicate the gap that we are filling by providing a brief review of the existing literature. We have also introduced a couple of theories that we believe provide a much-needed underpinning for this work.

Comment:

The content is not written in scientific. The objectives of this work need to be more focused into the analysis.

Response:

We apologize that we do not quite understand the comments here and what changes are being suggested by the reviewer.

Comment:

Result and discussion should be following independently.

Response:

Thank you for this. We have included a discussion section, as requested by the reviewer.

Comment:

The regression and multinomial logistic model didn’t explained proper scientific way and incomplete to proof the goals of the paper. Results are undependable as statistical results are merely stated without any analytical explanation.

Response:

We apologize that we do not quite understand the comments here and what changes are being suggested by the reviewer.

Comment:

Quantitative Analysis should be more in-depth, scientific and context specific for the specific area. And study is need for more information on policy implications of these particular findings.

Response:

Thank you. We have updated the conclusion section with policy recommendation derived more directly from the research findings.

Comment:

It uses inappropriate statistical tools for Table 3 & 4.

Response:

We apologize that we do not quite understand the comments here and what changes are being suggested by the reviewer.

---

## [Decision Letter · Decision Letter 1]

15 Dec 2020

PONE-D-20-22368R1

Mixed Methods Exploration of Ghanaian Women’s Domestic Work, Childcare and Effects on their Mental Health

PLOS ONE

Dear Dr. Lambon-Quayefio,

Thank you for submitting your manuscript to PLOS ONE. After careful consideration, we feel that it has merit but does not fully meet PLOS ONE’s publication criteria as it currently stands. Therefore, we invite you to submit a revised version of the manuscript that addresses the points raised during the review process.

ACADEMIC EDITOR: The reviewers still feel some minor revisions are required before recommending this paper for publication. Alongside the reviewer suggestions, I request authors to go for a professional English proof reading of the paper and also make the stylistic changes according to PLOS One author guidelines. 

We look forward to receiving your revised manuscript.

Kind regards,

Srinivas Goli, Ph.D.

Academic Editor

PLOS ONE

Additional Editor Comments (if provided):

The reviewers still feel some minor revisions are required before recommending this paper for publication. Alongside the reviewer suggestions, I request authors to go for a professional English proof reading of the paper and also make the stylistic changes according to PLOS One author guidelines.

Reviewers' comments:

Reviewer's Responses to Questions

**Comments to the Author**

1. If the authors have adequately addressed your comments raised in a previous round of review and you feel that this manuscript is now acceptable for publication, you may indicate that here to bypass the “Comments to the Author” section, enter your conflict of interest statement in the “Confidential to Editor” section, and submit your "Accept" recommendation.

Reviewer #1: All comments have been addressed

Reviewer #2: All comments have been addressed

Reviewer #3: All comments have been addressed

2. Is the manuscript technically sound, and do the data support the conclusions?

Reviewer #1: Yes

Reviewer #2: Partly

Reviewer #3: Yes

3. Has the statistical analysis been performed appropriately and rigorously? 

Reviewer #1: Yes

Reviewer #2: Yes

Reviewer #3: Yes

4. Have the authors made all data underlying the findings in their manuscript fully available?

Reviewer #1: No

Reviewer #2: Yes

Reviewer #3: Yes

5. Is the manuscript presented in an intelligible fashion and written in standard English?

Reviewer #1: Yes

Reviewer #2: No

Reviewer #3: Yes

6. Review Comments to the Author

Reviewer #1: The authors have addressed all the comments and revised version of the manuscript is up to the level of satisfaction. There are a few minor suggestions the authors may please consider.

1. In page no 7, the authors reported "snowballing sampling" as one of the sample selection procedures. It is suggested that the authors report it as "Snowball Sampling".

2. As per the information provided in paragraph 3 page number 8, it is unclear if the researchers used inductive approach to develop qualitative codes or deductive approach to do so. It would be better if authors could add a sentence or two clarifying it.

3. There is inconsistency in table numbers and the reporting of tables in the text. For example, table 1 comes two times (the first and second tables are numbered as table 1). Table 2 reported in paragraph 1 of descriptive results section (page 11) is not found. The authors may please check this.

4. The researchers may please check that the referencing is consistent and adheres to the standard format. For example; DOI was not reported in few references though DOI was available. Similarly, some references such as Teychenne at al. (2008) needs to be formatted.

Reviewer #2: Most of the comments are addressed and the current version of manuscript appears to have improved. However, there still are a few minor concerns especially, the language used in the manuscript needs substantia revision. Additionally, the discussion and conclusion section need a major revision.

Specific comments are as follows:

• The choice of word/phrase such as ‘sexual division of labour’ in page 3, ‘spending four times more effort’ in page 3, ‘wives’, ‘in order to end up with valid and well-substantiated conclusions’ in page 5, ‘five-level response’ in page 6 etc. to indicate a few, needs definite revision. Language consistency may be maintained.

• Considering the current version of the manuscript is rather long, it would be useful to use sub-heads throughout the manuscript.

• At certain places in the introduction, the authors have expressed their views, for instance in page 4 ‘Therefore, when women’s total domestic workload is high, the combination of different roles may actually damage her health. Furthermore, the influence of social economic status may moderate the relationship. On the one hand, women with higher socioeconomic status may experience lower mental distress if they are better able to afford domestic appliances and house helps to reduce time spent in domestic work. On the other hand, wealthier women who spend more time in domestic work may have poorer mental health outcomes if this increases the opportunity costs of earnings and/or leisure activities, among other reasons’. Such blanket statements can lose the focus of the argument. The authors need to use appropriate scientific reporting style.

• Use of italics may be avoided unless essential

• Referencing style needs to be consistent for eg. In page 5 (Adjei and Brand, 2018; Everard et al., 2000), Lawlor et al. (2002) is inappropriate.

• The manuscript is lengthy and commentary such as ‘The remainder of the paper is scheduled as follows: Section II describes the (qualitative and quantitative) data and empirical methodologies employed in this research. Section III presents results and discussions of research findings, while Section IV concludes with potential social and policy implications of this research’ in page 5 may be avoided.

• Details of data availability, funding and other support in the survey may be referred to the report if available or be placed in the supplementary file.

• Policy implications may be written in light of the findings and requires to be revised to tone down the ambitious claims made.

• Shouldn’t the results and discussion sections be separate?

• Although the manuscript has improved from the previous version but would need a careful copy edit before publication.

Reviewer #3: Overall performance of this manuscript is quiet interesting and suitable for the journal like 'PLOSE ONE'

7. PLOS authors have the option to publish the peer review history of their article (what does this mean?). If published, this will include your full peer review and any attached files.

Reviewer #1: No

Reviewer #2: **Yes: **Jayakant Singh

Reviewer #3: No

---

## [Author Response · Author response to Decision Letter 1]

19 Dec 2020

Comments from academic editor: 

The reviewers still feel some minor revisions are required before recommending this paper for publication. Alongside the reviewer suggestions, I request authors to go for a professional English proof reading of the paper and also make the stylistic changes according to PLOS One author guidelines. 

Response:

We are grateful for the reviewers’ comments and have made the suggested changes, as requested. We have also submitted the paper for proof reading and have adhered to the journal’s format.

Reviewer #1

The authors have addressed all the comments and revised version of the manuscript is up to the level of satisfaction. There are a few minor suggestions the authors may please consider.

Comment:

In page no 7, the authors reported "snowballing sampling" as one of the sample selection procedures. It is suggested that the authors report it as "Snowball Sampling".

Response:

This change has been made-thank you.

Comment:

As per the information provided in paragraph 3 page number 8, it is unclear if the researchers used inductive approach to develop qualitative codes or deductive approach to do so. It would be better if authors could add a sentence or two clarifying it.

Response:

An inductive approach was used, and this has been clarified in the text.

Comment:

There is inconsistency in table numbers and the reporting of tables in the text. For example, table 1 comes two times (the first and second tables are numbered as table 1). Table 2 reported in paragraph 1 of descriptive results section (page 11) is not found. The authors may please check this.

Response:

Thank you; this inconsistency in the table labels has been addressed.

Comment:

The researchers may please check that the referencing is consistent and adheres to the standard format. For example; DOI was not reported in few references though DOI was available. Similarly, some references such as Teychenne at al. (2008) needs to be formatted.

Response:

Thank you for these comments- these changes have been made.

Reviewer #2

Most of the comments are addressed and the current version of manuscript appears to have improved. However, there still are a few minor concerns especially, the language used in the manuscript needs substantial revision. Additionally, the discussion and conclusion section need a major revision.

Comment:

The choice of word/phrase such as ‘sexual division of labour’ in page 3, ‘spending four times more effort’ in page 3, ‘wives’, ‘in order to end up with valid and well-substantiated conclusions’ in page 5, ‘five-level response’ in page 6 etc. to indicate a few, needs definite revision. Language consistency may be maintained.

Response:

“Sexual division of labor” revised to “gendered division of labour”

“spending four times more effort” revised to “spend 4 times as much time”

“in order to end up with valid and well-substantiated conclusions” revised to “in order to draw valid conclusions”

“five-level response” maintained to indicate that each question on respondent’s emotional state expected to prompt one of 5 responses. 

Comment:

Considering the current version of the manuscript is rather long, it would be useful to use sub-heads throughout the manuscript.

Response:

This comment is well received, and this is what we sought to do with the last revision that was submitted:

The current structure is as follows:

I. Introduction

II. Data and Methodology

a. Quantitative Data

i. Selected variables of interest

b. Qualitative Data

c. Methods

III. Results 

a. Descriptive results

b. Results from regressions and qualitative data

i. Overburdened

ii. Physical effects

iii. Economic effects

c. Robustness checks for selection and reverse causality

IV. Discussion of results

V. Conclusion

We are happy to revise the current structure further as per the reviewer’s suggestions. 

Comment:

At certain places in the introduction, the authors have expressed their views, for instance in page 4 ‘Therefore, when women’s total domestic workload is high, the combination of different roles may actually damage her health. Furthermore, the influence of social economic status may moderate the relationship. On the one hand, women with higher socioeconomic status may experience lower mental distress if they are better able to afford domestic appliances and house helps to reduce time spent in domestic work. On the other hand, wealthier women who spend more time in domestic work may have poorer mental health outcomes if this increases the opportunity costs of earnings and/or leisure activities, among other reasons’. Such blanket statements can lose the focus of the argument. The authors need to use appropriate scientific reporting style.

Response:

Thank you for this comment. To clarify, the use of the word “may” is meant to present the authors’ hypotheses without asserting a firm stance before the scientific analyses are completed. Additionally, the presentation of two potentially dissenting effects of social status on mental health is meant to, again, to objectively explain that no conclusions can be drawn until these relationships are tested, as either outcome is possible.

Comment:

Use of italics may be avoided unless essential

Response:

Thank you- as per the reviewer’s comment, all italicized text has been revised.

Comment:

Referencing style needs to be consistent for eg. In page 5 (Adjei and Brand, 2018; Everard et al., 2000), Lawlor et al. (2002) is inappropriate.

Response:

Thank you- the referencing style has been checked for consistency.

Comment:

The manuscript is lengthy and commentary such as ‘The remainder of the paper is scheduled as follows: Section II describes the (qualitative and quantitative) data and empirical methodologies employed in this research. Section III presents results and discussions of research findings, while Section IV concludes with potential social and policy implications of this research’ in page 5 may be avoided.

Response:

Thank you- this section has been removed.

Comment:

Details of data availability, funding and other support in the survey may be referred to the report if available or be placed in the supplementary file.

Response:

While we had previously refrained from including this information in earlier versions of the paper, reviewer(s) disagreed. We are happy to defer to a universally agreed-upon position by the reviewers and editor in keeping or eliminating this from the main document.

Comment:

Policy implications may be written in light of the findings and requires to be revised to tone down the ambitious claims made.

Response:

The policy applications of the research, as stated in the conclusion, include:

1. Efforts should be made to encourage greater participation of men in domestic work. One way to do this is with early socialization of boys and girls

2. Importance of labour saving devices to reduce domestic work burdens that women face and subsequently, improve their mental health.

These prescriptions are not necessarily ambitious. We already find, from our qualitative surveys, that men are gradually willing to assist with domestic work, particularly childcare, where in the past, it was considered unthinkable. These recommendations are meant to encourage such behaviors and prescribe early socialization of both boys and girls as one way to achieve this.

We would like to highlight that the following prescription has been removed from the abstract “Additionally, family-friendly polices to reduce work-family incompatibilities that women face should be promoted. These may include publicly provided daycare centers and even nursing stations and childcare centers within workplaces.”

Comment:

Shouldn’t the results and discussion sections be separate?

Response:

Thank you- they are separate, and the headings have been revised to reflect this

Comment:

Although the manuscript has improved from the previous version but would need a careful copy edit before publication.

Response:

Thank you, this has been done.

Reviewer #3: 

Overall performance of this manuscript is quiet interesting and suitable for the journal like 'PLOSE ONE'

Response:

Thank you.

---

## [Editor Report · Decision Letter 2]

22 Dec 2020

Mixed Methods Exploration of Ghanaian Women’s Domestic Work, Childcare and Effects on their Mental Health

PONE-D-20-22368R2

Dear Dr. Lambon-Quayefio,

We’re pleased to inform you that your manuscript has been judged scientifically suitable for publication and will be formally accepted for publication once it meets all outstanding technical requirements.

Kind regards,

Srinivas Goli, Ph.D.

Academic Editor

PLOS ONE

Additional Editor Comments (optional):

This paper is now acceptable for publication if it meets the PLOS One authors' guidelines.
---

## [Editor Report · Acceptance letter]

22 Jan 2021

PONE-D-20-22368R2 

Mixed Methods Exploration of Ghanaian Women’s Domestic Work, Childcare and Effects on their Mental Health 

Dear Dr. Lambon-Quayefio:

I'm pleased to inform you that your manuscript has been deemed suitable for publication in PLOS ONE. Congratulations! Your manuscript is now with our production department. 

Kind regards, 

on behalf of

Dr. Srinivas Goli 

Academic Editor

PLOS ONE